# Validation of the *SavvyCheck*™ *Vaginal Yeast Test* for Screening Pregnant Women for Vulvovaginal Candidosis: A Prospective, Cross-Sectional Study

**DOI:** 10.3390/jof7030233

**Published:** 2021-03-20

**Authors:** Philipp Foessleitner, Herbert Kiss, Julia Deinsberger, Julia Ott, Lorenz Zierhut, Alex Farr

**Affiliations:** 1Department of Obstetrics and Gynecology, Division of Obstetrics and Feto-Maternal Medicine, Medical University of Vienna, A-1090 Vienna, Austria; philipp.foessleitner@meduniwien.ac.at (P.F.); herbert.kiss@meduniwien.ac.at (H.K.); n1547087@students.meduniwien.ac.at (J.O.); lorenz.zierhut@student.i-med.ac.at (L.Z.); 2Skin and Endothelium Research Division (SERD), Department of Dermatology, Medical University of Vienna, A-1090 Vienna, Austria; julia.deinsberger@meduniwien.ac.at

**Keywords:** antenatal care, *Candida*, point-of-care test, pregnancy, vulvovaginal candidosis

## Abstract

Pregnant women have an increased risk of vulvovaginal candidosis. Recurrent candidosis is under debate as a contributor to preterm birth, and vertical transmission may cause diaper dermatitis and oral thrush in the newborn. Apart from cultural methods, the gold standard for diagnosing candidosis is Gram staining, which is time-consuming and requires laboratory facilities. The objective of this prospective study was to validate a point-of-care vaginal yeast detection assay (*SavvyCheck*™ *Vaginal Yeast Test*) and to evaluate it in asymptomatic pregnant women. We enrolled 200 participants, 100 of whom had vulvovaginal candidosis according to Gram stain (study group) and 100 were healthy pregnant controls (control group). Of these, 22 participants (11%) had invalid test results. The point-of-care test of the remaining 85 and 93 study participants in the study and control groups, respectively, showed a sensitivity of 94.1%, specificity of 98.9%, positive predictive value of 90.3%, and negative predictive value of 99.4% when compared with Gram stain. In conclusion, we found a high correlation between the *SavvyCheck*™ *Vaginal Yeast Test* and Gram-stained smears during pregnancy. This suggests a potential role of this point-of-care test as a screening tool for asymptomatic pregnant women in early gestation.

## 1. Introduction

Vaginal infection in early pregnancy is associated with late miscarriage and spontaneous preterm birth (PTB), defined as birth prior to 37 gestational weeks [1]. Despite the many efforts made during recent decades, PTB remains the main cause of perinatal morbidity and mortality in industrialized countries [2]. It is well known that PTB is a multifactorial event, whose risk factors include previous PTB, advanced maternal age or high body mass index, nicotine abuse, various diseases, and infections [3,4], where infections are known to account for up to 40% of PTBs [5].

In addition to anaerobic bacteria, fungal diseases are under debate as contributors to PTB [6]. In particular, recurrent episodes of vulvovaginal candidosis (VVC) have been reported to increase the risk of PTB, which could be due to the continuous inflammatory stimulus during infection [7,8]. The prevalence of *Candida* during pregnancy has been reported to be at least 20% if colonization is examined by culture. The predominant species is *Candida albicans*, followed by non-*albicans* species that are often accompanied by milder symptoms and more likely to develop during pregnancy [9]. Of note, pregnancy itself increases the risk of acquiring VVC due to immunologic alterations, increased vaginal glycogen accumulation promoted by rising levels of estrogen, and reduced vaginal pH [9,10].

Early detection and treatment of potentially harmful pathogens at early gestation has proven to reduce the rate of PTB [11,12]. In our previous work, we reported an impressive reduction of PTBs through the integration of a screen-and-treat program in early pregnancy [11,12]. In a large randomized controlled trial, we were able to demonstrate that this risk reduction also applied for women with VVC apart from those who were diagnosed with bacterial vaginosis (BV) [11]. Moreover, it is well known that infants born to mothers with VVC frequently exhibit diaper dermatitis or oral thrush due to vertical mother-to-child transmission [13].

In clinical practice, diagnosis using potassium hydroxide preparation has proven to be inadequate due to a low sensitivity of 57.5% [14], and the use of culture methods to diagnose VVC is limited because of the relatively high costs and the time latency until the receipt of the report. Hence, Gram staining is widely considered the gold standard method for diagnosing vaginal infections [15]. However, microscopic evaluation of Gram-stained smears requires laboratory facilities, as well as trained and experienced staff, which are not widely available among medical facilities [16]. To overcome this problem, point-of-care tests have been introduced as an alternative that promises accurate and rapid diagnosis [17,18]. Point-of-care tests can be easily performed without additional equipment besides the test kit, and provide results within 10 min. One of these tests, the *SavvyCheck*™ *Vaginal Yeast Test*, has shown accuracy in symptomatic and non-pregnant women [17,18].

In the present study, we sought to validate this point-of-care vaginal yeast detection assay by evaluating its accuracy as a potential screening tool for asymptomatic VVC in early pregnancy.

## 2. Materials and Methods

### 2.1. Ethics Statement

This study has been approved by the Ethics Committee of the Medical University of Vienna (application number: 2115/2019). The study was conducted in accordance with the Declaration of Helsinki and Good Scientific Practice guidelines, following the STARD 2015 guidelines for validation studies [19]. Prior to study inclusion, all study participants signed an informed consent form. All patient records were pseudo-anonymized and de-identified prior to the analysis.

### 2.2. Setting and Study Population

This prospective, cross-sectional study was performed at the Medical University of Vienna, Department of Obstetrics and Gynecology, between 13 February 2020 and 12 January 2021. Our tertiary center serves about 2800 deliveries per year and is specialized in high-risk pregnancy care, including referrals from Central and Eastern Europe. All asymptomatic women who are registered for planned delivery undergo infection screening during prenatal consultation at early gestation as part of routine antenatal care. Women aged 18–50 years who had not received antibiotic treatment within the previous 2 weeks or any vaginal medication within the previous 72 h of their presentation were considered eligible for this study. By definition, study participants were not allowed to have any signs of conspicuous redness, discharge, or vaginal itching.

### 2.3. Sampling and Gram-Staining Procedure

Vaginal smears were obtained by vaginal fluid collection with sterile Dacron-tipped swabs (CLASSIQ Swabs™, Copan Italia S.p.A., Brescia, Italy) from the lateral vaginal wall and posterior fornix vaginae. Vaginal discharge from the first of two swabs was applied to a microscope slide and Gram-stained. The second of the two swabs was stored in a sterile, single-use plastic container and used for the point-of-care test, depending on the results of Gram staining. Gram-stained smears were microscopically analyzed by one of five biomedical laboratory assistants, who were trained and experienced in gynecological cytopathology, at a laboratory certified according to DIN EN ISO 9001:2008. Vaginal microbiota were classified according to Nugent et al. [20], thus, a score of 0–3 was considered normal, 4–6 as dysbiosis, and 7–10 as BV. Additionally, the presence of *Candida* species and/or *Trichomonas vaginalis* was evaluated microscopically. The participant was considered VVC-positive in case of presence of hyphae in the specimen, using binocular microscopic vision at 100-times magnification. The participant was considered VVC-negative in case of a total absence of oidia and hyphae in the specimen. If BV was diagnosed, participants were treated with 2% clindamycin vaginal cream for 6 days in case of primary disease, or 0.3 g oral clindamycin twice a day for 7 days in case of recurrent disease. In case of VVC, treatment with 0.1 g clotrimazole vaginal cream for 6 days was initiated. Participants with trichomoniasis were prescribed 0.5 g metronidazole vaginal cream for 7 days [21]. Antibiotic treatment was followed by the application of lactobacilli-containing vaginal capsules for 6 days in order to support rebuilding of the vaginal microbiota [22].

### 2.4. Point-of-Care Testing Procedure

The *SavvyCheck*™ *Vaginal Yeast Test* (Savyon Diagnostics, Ashdod, Israel) was used as recommended by the manufacturer (Figure 1). This assay uses the concept of a lateral flow immunoassay system; the swab is therefore mixed with an extraction buffer liquid placed in a cap situated at the proximal end of the device for 20 s. To convey the liquid onto the test strip, the cap of the device is rotated 2 times. The probe then flows by capillary forces along the various strip components. The first immunologic interaction develops between the extracted yeast antigen and the anti-*Candida* polyclonal antibody conjugated to a colored bead, generating a colored antibody–antigen complex. The now newly assembled complex migrates to a second anti-*Candida* polyclonal antibody, which is adherent to the membrane at the test line. This second immunologic interaction generates a visual signal along the strip due to the formation of concentrated colored tags at this specific location. After 10 min, a blue line appears in the control region (C), confirming that the assay was performed correctly. At a minimal fungal load of 3 × 10^3^, a blue line emerges in the test region (T), and the test can therefore be considered positive for VVC. In the absence of VVC, there is no line in the test region, and the test result is considered negative.

### 2.5. Study Groups

Women with microscopic evidence of VVC on Gram-stained smears (i.e., presence of hyphae) were assigned to the study group, whereas those without evidence of VVC (i.e., neither oidia nor hyphae) were assigned to the control group. Immediately after the Gram-staining procedure, the *SavvyCheck*™ *Vaginal Yeast Test* was performed by trained healthcare personnel as previously described. All demographic data were derived from obstetric databases and patient charts using the PIA Fetal Database, version 5.6.28.56 (General Electric Company, GE Viewpoint, Munich, Germany). The inclusion criteria for this study are shown in Figure 2.

### 2.6. Statistical Analysis

Statistical analysis was performed using SPSS Statistics version 27.0 (IBM, Armonk, NY, USA). Graphs and figures were drawn using GraphPad Prism (GraphPad Software, San Diego, CA, USA) and Lucidchart (Lucid Software Inc., South Jordan, UT, USA). Descriptive statistics were used to summarize demographic information. Continuous variables are presented as mean (±standard deviation); ordinally scaled variables are presented as median [interquartile range] or number (percentage); binary variables are presented as numbers (percentages). Sensitivity, specificity, positive predictive value (PPV), and negative predictive value (NPV) were calculated for the *SavvyCheck*™ *Vaginal Yeast Test*. To calculate PPV and NPV, we assumed a VVC prevalence of 9.8%, according to the existing literature [12].

## 3. Results

We enrolled 200 pregnant and asymptomatic women. Of these, 100 participants were assigned to the study group, and 100 participants were assigned to the control group. A total of 22 participants (15 in the study group and 7 in the control group) were excluded from the study because of an invalid test result, showing no line in the control region (C) of the point-of-care test. Consequently, statistical analyses were performed on the remaining 178 participants.

The mean maternal age of both groups at vaginal sampling was 32.2 (±5.6) years. The mean gestational age was 19.3 (±7.6) weeks in the study group and 17.6 (±8.0) weeks in the control group. In the study group, 57 (67.1%) of the women had normal vaginal microbiota, 25 (29.4%) presented with dysbiosis, and 3 (3.5%) were diagnosed with BV in addition to VVC. All 93 participants in the control group showed normal microbiota on Gram staining. The maternal characteristics are shown in Table 1.

When analyzing the results of the *SavvyCheck*™ *Vaginal Yeast Test*, we found that 80 of the 85 participants in the study group were correctly diagnosed with VVC, accounting for a sensitivity of 94.1%. In addition, 92 of the 93 participants in the control group correctly tested as negative, resulting in a specificity of 98.9%. Consequently, we calculated a PPV and NPV of 90.3% and 99.4%, respectively, for the *SavvyCheck*™ *Vaginal Yeast Test* (Table 2). The test results and vaginal microbiota of the participants in the study group are shown in Figure 3.

## 4. Discussion

Women are at increased risk of acquiring VVC during pregnancy [9,23]. An easy-to-use point-of-care test could screen for VVC in the absence of cost-intensive medical facilities [16]. To the best of our knowledge, our study is the first to evaluate such a test in pregnant women and validate its results with those on Gram stain. We found that the *SavvyCheck*™ *Vaginal Yeast Test* has a high sensitivity, specificity, PPV, and NPV.

Vaginal yeast detection assays have been available for the past two decades and have proven to be effective in diagnosing VVC in symptomatic and non-pregnant individuals [17,18,24]. These point-of-care tests are easily accessible over-the-counter and cheaper than culture methods. This, in turn, could reduce unnecessary patient consultations, diagnostic procedures, and unnecessary treatments, thereby reducing healthcare costs [17,18]. In addition, these tests offer fast results when compared to conventional culture methods, Gram staining, or DNA hybridization tests [25]. On a practical level, point-of-care tests could offer the possibility of screening women by healthcare personnel in the absence of cost-intensive medical facilities, or even by the women themselves.

The *SavvyCheck*™ *Vaginal Yeast Test* was previously validated for use in non-pregnant women [17,18]. Compared to Gram staining, the available literature suggests a sensitivity of 93% for symptomatic women, as well as a specificity of 95%, PPV of 89%, and NPV of 97% [18]. Studies that compared the test with cultural methods reported a sensitivity and specificity of 77–79% and 76–96%, respectively, as well as a PPV and NPV ranging from 72–94% and 81–89%, respectively [17,18].

In this study, we aimed to validate this test for asymptomatic pregnant women, which is clearly different from previously published work. Sensitivity and specificity are of particular importance in symptomatic individuals for the detection or exclusion of the disease as the source of symptoms; however, PPV and NPV are even more relevant in our study setting, as we screened asymptomatic individuals for the disease [26,27]. The PPV of 90.3% and NPV of 99.4% suggested adequate accuracy in VVC screening during pregnancy. Of note, the relatively high no-show rate of 11% in our study was likely attributed to an underlying technical problem with the device, as the fluid was not conveyed onto the test strip.

This attempt to find an accurate and easy-to-use screening tool for VVC follows the results of our previous work, showing a significant reduction of PTB through early infection screening in asymptomatic pregnant women [11,12], as well as an increased risk for PTB among women with recurrent VVC [7]. We are aware that the latter is still under debate [7,8]. Two cohort studies from the 1990s could not find a significant association between PTB and moderate-to-heavy fungal growth [8,28,29], whereas a longitudinal study from a large population-based dataset of the Hungarian Case-Control Surveillance of Congenital Abnormalities, including more than 38,000 newborns, reported that treatment with topical clotrimazole was associated with a 34–64% reduction in the prevalence of PTB [30,31]. Another retrospective study from the USA observed a 49% reduction in PTB after women were treated with vaginally applied azoles for *Candida* vaginitis [8,32]. Our own work, as well as that of Roberts et al., supports the idea that VVC during pregnancy somehow increases the risk of PTB, at least in cases with recurrent disease [7,8].

This study has several limitations, including the lack of cultural methods. We chose this procedure for the sake of cost reduction, and as we know, Gram stain is reliable for diagnosing candidosis under the microscope. Moreover, *Candida albicans* is a commensal fungus that is able to colonize the female genital tract without transiting into a pathogen that causes VVC [9]; therefore, screening asymptomatic women might also be controversial. However, we considered this procedure reasonable, as this test requires a certain fungal load for a positive test result. The strengths of our study include the large study population compared to the available literature, and the study setting that we used, which ensured accurate diagnosis and homogeneous antenatal care.

## 5. Conclusions

Our study demonstrates that the vaginal yeast detection assay *SavvyCheck*™ *Vaginal Yeast Test* correlates with Gram stain in asymptomatic women during pregnancy. We therefore consider this point-of-care test as a reliable and easy-to-use alternative for diagnosing VVC during pregnancy, which also suggests a role as a screening tool for pregnant women. Similar approaches are warranted for other hazardous pathogens that potentially increase the risk for preterm birth.

## Figures and Tables

**Figure 1 jof-07-00233-f001:**
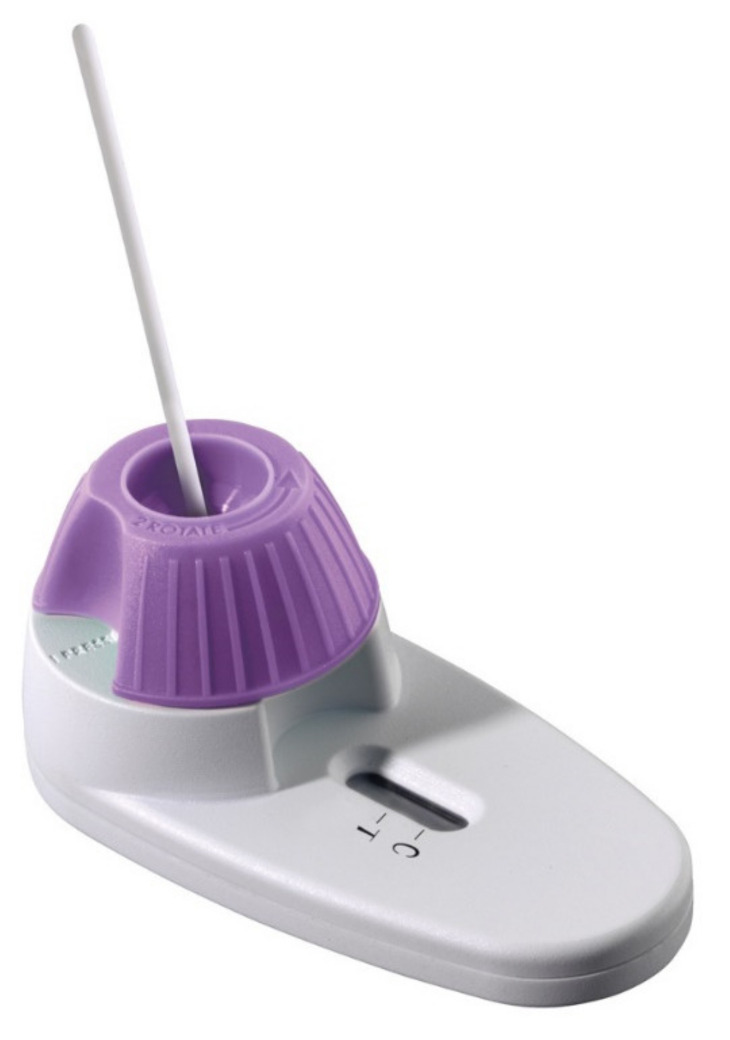
*SavvyCheck*™ *Vaginal Yeast Test* kit (picture used with permission of the manufacturer).

**Figure 2 jof-07-00233-f002:**
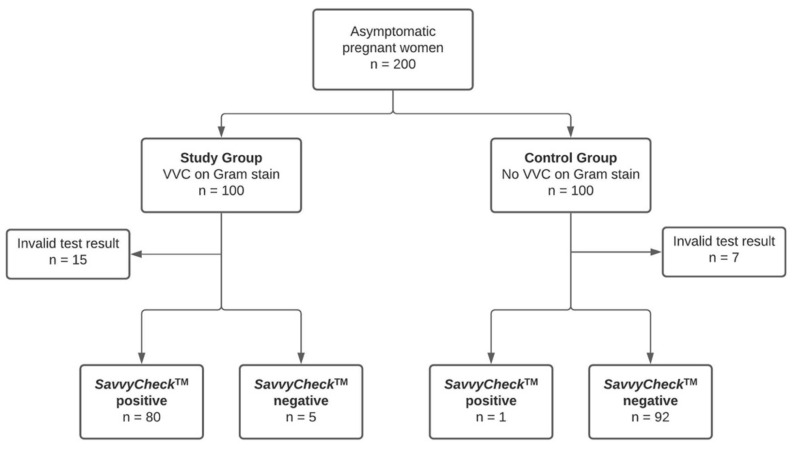
Criteria of inclusion of 200 enrolled, asymptomatic pregnant women who were screened for vulvovaginal candidosis (VVC) using Gram-stained smears and the *SavvyCheck*™ *Vaginal Yeast Test*.

**Figure 3 jof-07-00233-f003:**
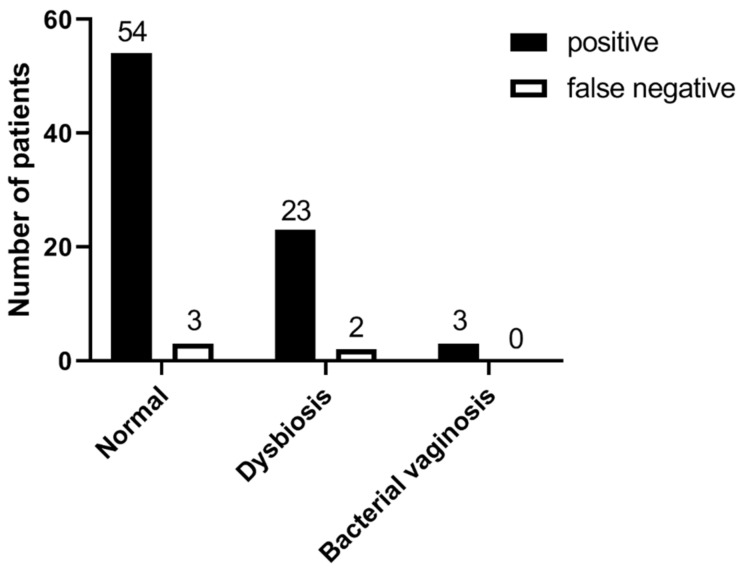
Results of the *SavvyCheck*™ *Vaginal Yeast Test* and vaginal microbiota in 85 study participants of the study group.

**Table 1 jof-07-00233-t001:** Characteristics of 178 asymptomatic pregnant women who were screened for VVC using. Gram-stained smears and the *SavvyCheck*™ *Vaginal Yeast Test*.

	Study Group (n = 85)	Control Group (n = 93)	All (n = 178)
Maternal Age	32.2 (±5.6)	32.2 (±5.6)	32.2 (±5.6)
Gravidity	2 (1–8)	3 (1–13)	2 (1–13)
Parity	1 (0–5)	1 (0–6)	1 (0–6)
**Smoking**			
Yes	11 (12.9%)	13 (14%)	24 (13.5%)
No	74 (87.1%)	80 (86%)	154 (86.5%)
**Previous preterm birth**			
Yes	6 (7.1%)	13 (14%)	19 (10.7%)
No	79 (92.9%)	80 (86.0%)	159 (89.3%)
Gestational week at screening	19.3 (±7.6)	17.6 (±8.0)	18.4 (±7.9)
**Vaginal microbiota**			
Normal	57 (67.1%)	93 (100%)	150 (84.3%)
Dysbiosis	25 (29.4%)	0 (0%)	25 (14%)
Bacterial vaginosis	3 (3.5%)	0 (0%)	3 (1.7%)
**Vulvovaginal candidosis**			
Yes	85 (100%)	0 (0%)	85 (47.8%)
No	0 (0%)	93 (100%)	93 (52.2%)

Data are presented as number (percentage), mean (±standard deviation), or median (range).

**Table 2 jof-07-00233-t002:** Validation of the *SavvyCheck*™ *Vaginal Yeast Test* in comparison to Gram-stained smears in a total of 178 asymptomatic pregnant women who were screened for VVC.

*SavvyCheck*™ *Vaginal Yeast Test*	Study Group (n = 85)	Control Group (n = 93)	Total (n = 178)
Positive	80	1	81
Negative	5	92	97
Total	85	93	178

Sensitivity, 94.1%; specificity, 98.9%; positive predictive value, 90.3%; negative predictive value, 99.4% (assumed VVC prevalence, 9.8%) [12].

## Data Availability

The data are available on request to the corresponding author.

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
