# Peer review of "Validation of the SavvyCheckVaginal Yeast Test for Screening Pregnant Women for Vulvovaginal Candidosis: A Prospective, Cross-Sectional Study"

_jof, 2021, doi:10.3390/jof7030233_

Round 1

Reviewer 1 Report

The submitted paper describes validation of the SavvycheckTM Vaginal Yeast Test for screening pregnant women for vulvovaginal candidosis. In pregnancy, prompt and accurate diagnosis of vulvovaginal candidosis is of great significance in prevention of premature rupture of membranes, preterm labor, chorioamnionitis, late miscarriage and congenital cutaneous candidiasis. This study provides the first evaluation of rapid, easy-to-perform, SavvycheckTM Vaginal Yeast Test in asymptomatic pregnant women and compares obtained results with Gram stain as standard. Based on obtained results, this point-of-care test may represent a part of screening programs and strategies to reduce the risk of pregnancy complications. The manuscript is written very clearly, easy to read and to understand, and therefore deserves to be published in “Journal of Fungi”. This reviewer has only a few minor suggestions:

  1. Page 2, line 39 - Please, include immunologic alterations, increased vaginal glycogen accumulation promoted by rising levels of estrogen and reduced vaginal pH as risk factors for vulvovaginal candidosis in pregnancy
  2. Please, add a few sentences about the prevalence of vulvovaginal candidosis and describe the most frequently isolated Candida species as causative agents of this infection in pregnant women

Author Response

REVIEWER 1

Reviewer 1, Comment 1:

The submitted paper describes validation of the SavvycheckTM Vaginal Yeast Test for screening pregnant women for vulvovaginal candidosis. In pregnancy, prompt and accurate diagnosis of vulvovaginal candidosis is of great significance in prevention of premature rupture of membranes, preterm labor, chorioamnionitis, late miscarriage and congenital cutaneous candidiasis. This study provides the first evaluation of rapid, easy-to-perform, SavvycheckTM Vaginal Yeast Test in asymptomatic pregnant women and compares obtained results with Gram stain as standard. Based on obtained results, this point-of-care test may represent a part of screening programs and strategies to reduce the risk of pregnancy complications. The manuscript is written very clearly, easy to read and to understand, and therefore deserves to be published in “Journal of Fungi”. This reviewer has only a few minor suggestions.

Response to Reviewer 1, Comment 1:

We appreciate the helpful comments of Reviewer 1 that have been addressed accordingly in the revised version of our manuscript.

Reviewer 1, Comment 2:

Page 2, line 39 - Please, include immunologic alterations, increased vaginal glycogen accumulation promoted by rising levels of estrogen and reduced vaginal pH as risk factors for vulvovaginal candidosis in pregnancy

Response to Reviewer 1, Comment 2:

We thank you for this important comment. As suggested, we have now revised this paragraph, shortly pointing out the risk factors for VVC as mentioned by the reviewer (lines 42–45).

Reviewer 1, Comment 3:

Please, add a few sentences about the prevalence of vulvovaginal candidosis and describe the most frequently isolated Candida species as causative agents of this infection in pregnant women

Response to Reviewer 1, Comment 3:

This is indeed a very important point. As suggested, we have now included the following paragraph in the introduction section of our revised manuscript: “The prevalence of Candida during pregnancy has been reported to be at least 20% if colonization is examined by culture. The predominant species is Candida albicans, followed by non-albicans species that are often accompanied by milder symptoms and more likely to develop during pregnancy.” (lines 39–42).

Reviewer 2 Report

In the present manuscript the authors validate the point-of-care vaginal yeast detection assay SavvyCheck™ Vaginal Yeast Test in a cohort of 200 participants and compare it to the well-known and establish Gram staining method for diagnosis of vulvovaginal candidiasis. Although the method has already been described in previous trials, the authors report for the first time its application in asymptomatic pregnant women. VVC affects up to 75% of women of childbearing age worldwide. Among the host-specific factors, hormonal cycle and pregnancy play a key role in VVC onset and association in symptomatic women with preterm is still under debate. Even though the majority of the individuals report the classical symptoms, a minority of women are asymptomatically colonised thus resulting in inaccurate diagnosis and subsequent reduced quality of life.

This study provides novelty and applicability in both the general medical practice and the clinical research by expanding the application of a fast and non-invasive screening method.

The research question is well defined and the data largely support the main objective and claims. The logical flow and the structure of the narrative are clear and comprehensive as well.

I recommend publication of the manuscript after some minor changes as it will be of great interest for the readership of Journal of Fungi and perfectly suitable for the section Fungal Infections in Obstetrics and Gynecology.

Minor suggestions:

  1. The authors state in line 19-20 , 154-,and 211-212: “Of these, 22 participants (11%) had invalid test results”. Can the authors speculate more in the discussion part, in lines 211-212. Something related to test method, or the low fungal load of the individual?
  2. Why hasn’t the test being repeated in those individuals ?
    Did the Gram staining revealed any link or discrepancy with the invalid test in those 11%?
  3. I suggest to also cite the KOH method for diagnosis in the introduction part.
  4. Technical correction: in line 103 remove the double space.
  5. Line 112 correct the typo or make it more clear: “the cap twice is rotated 2 times”
  6. Specify in the materials and methods section the fungal load necessary for a positive test result.
  7. Technical correction: line spacing in lines 235-240 needs to be adjusted to 1.5.

Author Response

REVIEWER 2

Reviewer 2, Comment 1:

The authors state in line 19-20, 154-, and 211-212: “Of these, 22 participants (11%) had invalid test results”. Can the authors speculate more in the discussion part, in lines 211-212. Something related to test method, or the low fungal load of the individual?

Response to Reviewer 2, Comment 1:

Thank you for your valuable comment. The invalid test results occurred due to technical issues in the test kits. In fact, the fluid was not conveyed onto the test strip, but stuck inside the cap. We have already informed the manufacturer about this problem and added this additional information in the Discussion section of our revised manuscript (lines 221–223).

Reviewer 2, Comment 2:

Why hasn’t the test being repeated in those individuals?

Did the Gram staining revealed any link or discrepancy with the invalid test in those 11%?

Response to Reviewer 2, Comment 2:

Thank you for this question. Unfortunately, we were unable to repeat the test in these participants, as two swabs were taken during the screening and examination procedure: one for Gram staining, and another one for the Savvycheck™ Vaginal Yeast Test. Regarding your second question in this comment: Gram staining did not reveal any link or discrepancy with the invalid results in these women. As mentioned in our response to Reviewer 2, Comment 1, invalid tests occurred due to a technical issue with the test kit.

Reviewer 2, Comment 3:

I suggest to also cite the KOH method for diagnosis in the introduction part.

Response to Reviewer 2, Comment 3:

We thank Reviewer 1 for this suggestion. We have now added a paragraph about potassium hydroxide preparation for diagnostics in the introduction section of our revised manuscript (lines 54–58): “In clinical practice, diagnosis using potassium hydroxide preparation has proven to be inadequate due to a low sensitivity of 57.5%, and the use of culture methods to diagnose VVC is limited because of the relatively high costs and the time latency until the receipt of the report.”

Reviewer 2, Comment 4:

Technical correction: in line 103 remove the double space.

Response to Reviewer 2, Comment 4:

Thank you for this suggestion, which has been changed accordingly.

Reviewer 2, Comment 5:

Line 112 correct the typo or make it more clear: “the cap twice is rotated 2 times”

Response to Reviewer 2, Comment 5:

Thank you for this suggestion, which has been changed accordingly.

Reviewer 2, Comment 6:

Specify in the materials and methods section the fungal load necessary for a positive test result.

Response to Reviewer 2, Comment 6:

Thank you for this important comment. Die minimal fungal load for detection is 3 x 103. We have now added this to the methods section of our revised manuscript (line 126–127).

Reviewer 2, Comment 7:

Technical correction: line spacing in lines 235-240 needs to be adjusted to 1.5.

Response to Reviewer 2, Comment 7:

The line spacing has been adapted as suggested.

Reviewer 3 Report

The study from Foessleitner et. al provides a prospective evaluation of a commercial Candida detection test as a screening test for asymptomatic pregnant women in early gestation.  The study design is well structured and the population size adequate for drawing valid conclusions. However, there are few minor points that need to be addressed in the text from the authors:

1) Did the authors consider repeating the test in the n=15 VVC and n=7 non-VVC invalid test results? Was the invalid results a problem of the kit itself, or of the procedure/swabbing?

2) How much time passed between the cytopathological diagnosis and the test with SavvyCheck test? How were the patient swabs stored in the meanwhile? This should be specified in the methods as well as the kind of sterile swabs (e.g. flocked, cotton...)

3) Who performed the Savvy Check test? Was this done by healthcare personnel "blindly"before knowing the result of the cytopathologist or was it stored and then performed at a later date? 

4) Why was the SavvyCheck test chosen instead of other commercial kits available from other brands? 

5) How would the author on a practical level implement this kit as a screening test? Via the GPs? In the hospital directly or by asking women directly to perform such a test?

6) Assuming this screening practice is implemented in the future. Wouldn't this risk underestimating/losing the number of cases of bacterial vaginosis that are also responsible for premature birth? Have the authors not considered implementing in this screening a rapid kit for bacterial or Trichomonas vaginal infections of equal effectiveness (if any) in diagnosing asymptomatic non-fungal vaginosis in parallel? The author should discuss this.

Author Response

REVIEWER 3

Reviewer 3, Comment 1:

Did the authors consider repeating the test in the n=15 VVC and n=7 non-VVC invalid test results? Was the invalid results a problem of the kit itself, or of the procedure/swabbing?

Response to Reviewer 3, Comment 1:

We thank Reviewer 2 for this valuable comment. The invalid test results likely occurred due to a technical problem with the test kit. Please see our responses to Reviewer 2, Comments 1 and 2. We have also added this information to the discussion section of the revised version of our manuscript (lines 221–223).

Reviewer 3, Comment 2:

How much time passed between the cytopathological diagnosis and the test with SavvyCheck test? How were the patient swabs stored in the meanwhile? This should be specified in the methods as well as the kind of sterile swabs (e.g. flocked, cotton...).

Response to Reviewer 3, Comment 2:

Thank you for this comment. The Savvycheck™ Vaginal Yeast Test was performed immediately after the Gram stain result was available; the point-of-care test was subsequently performed within a maximum of two hours after sampling. The swabs were stored in a sterile, single-use plastic container in the meantime. Sterile Dacron tipped swabs were used for sampling (Copan Italia S.p.A., Brescia, Italy). We added this information to the methods section of our revised manuscript (lines 92–94, 95–97 and 136–138).

Reviewer 3, Comment 3:

Who performed the Savvy Check test? Was this done by healthcare personnel "blindly" before knowing the result of the cytopathologist or was it stored and then performed at a later date?

Response to Reviewer 3, Comment 3:

Thank you for this question. The Savvycheck™ Vaginal Yeast Test was performed unblinded by healthcare personnel immediately after Gram staining. The healthcare personnel that performed the test had been extensively trained in the handling of this text prior to the initiation of the study. Temporary storage of the swabs that were used for the test is described in our response to Reviewer 3, Comment 2. We have now included this information in the methods section of our revised manuscript (lines 136–138).

Reviewer 3, Comment 4:

Why was the SavvyCheck test chosen instead of other commercial kits available from other brands?

Response to Reviewer 3, Comment 4:

We appreciate this question. When choosing an eligible point-of-care test for our study, we identified the Savvycheck™ Vaginal Yeast Test as the most accurate detection assay for symptomatic women with VVC. As we were seeking for an effective method to screen asymptomatic pregnant women at early gestation, this test was the only one that fulfilled our requirement and had not yet been used in pregnant women.1, 2

Reviewer 3, Comment 5:

How would the author on a practical level implement this kit as a screening test? Via the GPs? In the hospital directly or by asking women directly to perform such a test?

Response to Reviewer 3, Comment 5:

Thank you for this comment. On a practical level, we believe that point-of-care tests generally offer the possibility to screen women in the absence of cost-intensive medical facilities, e.g. at gynecologic/GPs’ offices or hospitals, where cytopathological examination is not available. Apart from that, self-sampling and consecutive testing by each individual woman could also be a theoretical option to screen. We have now included the following paragraph in the discussion part of our revised manuscript (lines 206–208): “On a practical level, point-of-care tests could offer the possibility of screening women by healthcare personnel in the absence of cost-intensive medical facilities, or even by the women themselves.” We believe that this important point should now be clear for the readers.

Reviewer 3, Comment 6:

Assuming this screening practice is implemented in the future. Wouldn't this risk underestimating/losing the number of cases of bacterial vaginosis that are also responsible for premature birth? Have the authors not considered implementing in this screening a rapid kit for bacterial or Trichomonas vaginal infections of equal effectiveness (if any) in diagnosing asymptomatic non-fungal vaginosis in parallel? The author should discuss this.

Response to Reviewer 3, Comment 6:

We thank you for this important question. We are aware of the role of bacterial vaginosis as causative factor for preterm birth, as it has widely been described.3 As a consequence, screening for other potentially hazardous pathogens should also be performed. Point-of-care screening tests for the detection of bacterial vaginosis and/or trichomoniasis are commonly available on the market and will be evaluated in one of our upcoming studies. We have included this point in the conclusion section of our revised manuscript (lines 247–252).
